# Identification and analysis of individuals who deviate from their genetically-predicted phenotype

**Gareth Hawkes**[1], **Loic Yengo**[2], **Sailaja Vedantam**[3], **Eirini Marouli**[4], **Robin N. Beaumont**[1], **the GIANT Consortium**, **Jessica Tyrrell**[1], **Michael N. Weedon**[1], **Joel Hirschhorn**[5], **Timothy M. Frayling**[1]*, **Andrew R. Wood**[1]*

**1** Genetics of Complex Traits, College of Medicine and Health, University of Exeter, Exeter, Devon, United Kingdom, **2** Institute for Molecular Bioscience, The University of Queensland, Brisbane, Australia, **3** Endocrinology, Boston Children's Hospital, Sharon, Massachusetts, United States of America, **4** William Harvey Research Institute, Barts and The London School of Medicine and Dentistry Queen Mary University of London, London, United Kingdom, **5** Boston Children's Hospital/Broad Institute, Boston, Massachusetts, United States of America

* T.M.Frayling@exeter.ac.uk (TMF); A.R.Wood@exeter.ac.uk (ARW)

**Data Availability Statement:** The research utilised data from the UK Biobank resource carried out under UK Biobank application number 9072. UK Biobank protocols were approved by the National

## Abstract

Findings from genome-wide association studies have facilitated the generation of genetic predictors for many common human phenotypes. Stratifying individuals misaligned to a genetic predictor based on common variants may be important for follow-up studies that aim to identify alternative causal factors. Using genome-wide imputed genetic data, we aimed to classify 158,951 unrelated individuals from the UK Biobank as either concordant or deviating from two well-measured phenotypes. We first applied our methods to standing height: our primary analysis classified 244 individuals (0.15%) as misaligned to their genetically predicted height. We show that these individuals are enriched for self-reporting being shorter or taller than average at age 10, diagnosed congenital malformations, and rare loss-of-function variants in genes previously catalogued as causal for growth disorders. Secondly, we apply our methods to LDL cholesterol (LDL-C). We classified 156 (0.12%) individuals as misaligned to their genetically predicted LDL-C and show that these individuals were enriched for both clinically actionable cardiovascular risk factors and rare genetic variants in genes previously shown to be involved in metabolic processes. Individuals whose LDL-C was higher than expected based on the genetic predictor were also at higher risk of developing coronary artery disease and type-two diabetes, even after adjustment for measured LDL-C, BMI and age, suggesting upward deviation from genetically predicted LDL-C is indicative of generally poor health. Our results remained broadly consistent when performing sensitivity analysis based on a variety of parametric and non-parametric methods to define individuals deviating from polygenic expectation. Our analyses demonstrate the potential importance of quantitatively identifying individuals for further follow-up based on deviation from genetic predictions.

Research Ethics Service Committee. Individual-level data cannot be shared publicly because of data access policies of the UK Biobank. Data are available from the UK Biobank for researchers who meet the criteria for access to datasets to UK Biobank (http://www.ukbiobank.ac.uk). The weights used to calculate the polygenic score for height is available in Table C in S1 Data. The weights used to calculate the polygenic score for LDL-cholesterol, calculated in a meta-analysis excluding UK Biobank, are available from the Global Lipids Genetics Consortium at https://csg. sph.umich.edu/willer/public/glgc-lipids2021/.

**Funding:** This manuscript is part of the Stratification of Obesity Phenotypes to Optimize Future Obesity Therapy (SOPHIA) project. SOPHIA has received funding from the Innovative Medicines Initiative 2 Joint Undertaking under grant agreement No. 875534. This Joint Undertaking support from the European Union's Horizon 2020 research and innovation program and EFPIA and T1D Exchange, JDRF, and Obesity Action Coalition (www.imisophia.eu). GH has received a salary from the Innovative Medicines Initiative 2 Joint Undertaking under grant agreement No 875534. JT is supported by an Academy of Medical Sciences (AMS) Springboard award, which is supported by the AMS, the Wellcome Trust, GCRF, the Government Department of Business, Energy and Industrial strategy, the British Heart Foundation and Diabetes UK [SBF004\1079]. ARW is supported by the Academy of Medical Sciences / the Wellcome Trust / the Government Department of Business, Energy and Industrial Strategy / the British Heart Foundation / Diabetes UK Springboard Award [SBF006\1134]. TMF is supported by MRC awards MR/WO14548/1 and MR/T002239/1. LY is supported the Australian Research Council (DE200100425). JNH and SV are supported by R01 DK075787. The funders had no role in study design, data collection and analysis, decision to publish, or preparation of the manuscript.

**Competing interests:** The authors have declared that no competing interests exist.

## Author summary

Human genetics is becoming increasingly useful to help predict human traits across a population owing to findings from large-scale genetic association studies and advances in the power of genetic predictors. This provides an opportunity to potentially identify individuals that deviate from genetic predictions for a common phenotype under investigation. For example, an individual may be genetically predicted to be tall, but be shorter than expected. It is potentially important to identify individuals who deviate from genetic predictions as this can facilitate further follow-up to assess likely causes. Using 158,951 unrelated individuals from the UK Biobank, with height and LDL cholesterol as exemplar traits, we demonstrate that approximately 0.15% and 0.12% of individuals deviate from their genetically predicted phenotypes, respectively. We observed these individuals to be enriched for a range of rare clinical diagnoses, as well as rare genetic factors that may be causal. Our analyses also demonstrate several methods for detecting individuals who deviate from genetic predictions that can be applied to a range of continuous human phenotypes.

## Introduction

Since 2007 [1], genome-wide association studies (GWAS) have identified thousands of associations between common single nucleotide polymorphisms (SNPs) and human traits. This has resulted in an increase in the variance explained and out-of-sample prediction accuracy for common human traits [2–4]. For example, the largest published GWAS meta-analysis for height identified 12,111 SNP-associations that explained $\sim 40\%$ of the variance in height among individuals of European genetic ancestry and between 10–20% in other genetic ancestries [3]. Although the amount of variance explained for common quantitative traits continues to increase, less is understood of how common genetic variation contributes to phenotypic variation in the extreme tails of quantitative trait distributions [5], and whether individuals who present relatively extreme deviation from their expected phenotype given their common SNP-based predictor can be identified.

It may be important to identify individuals who deviate from their predicted phenotype based on an assumed polygenic model of association because they may be more likely to carry rarer and more penetrant pathogenic mutations or have some other cause to their phenotype. Specific alternative causes of an extreme phenotype may require targeted clinical investigations for an individual.

Using height and LDL cholesterol (LDL-C) as exemplar traits, chosen for their high heritability and clinical relevance respectively, we aimed to classify individuals who deviate from their genetically predicted phenotype, using 158,951 unrelated individuals from the UK Biobank with whole exome-sequencing data. We subsequently aimed to determine if individuals classified as misaligned to their genetically predicted height were enriched for recall of being relatively short or tall in childhood, disproportionate body stature, clinical diagnoses of syndromes associated with extreme stature, carriers for rare genetic variation relevant to height, or environmental factors that may have influenced growth. Secondly, we aimed to determine if individuals classified as misaligned to their genetically predicted LDL-C were at higher risk of heart disease, more or less likely to have type 2 diabetes, or were carriers for rare genetic variation relevant to LDL-C. Finally, we assessed the sensitivity of our results based on four methods, each with two thresholds, that have the potential to be used to identify individuals whose phenotype deviates from the expectation based on their polygenic score.

## Results

### Standing height

**A derived polygenic score for height explains 32% of the variance in the UK Biobank.**
We derived a polygenic score using conditional effect estimates of 3,198 SNPs reaching $P < 5 \times 10^{-8}$ obtained from a meta-analysis of 1.2M individuals from European-based studies (excluding the UK Biobank) contributing to the Genetic Investigation of ANthropometric Traits (GIANT) consortium. The polygenic score explained 31.6% of the variance in height among 158,951 unrelated individuals of European genetic ancestry with exome sequencing in the UK Biobank (Fig 1). A 1SD increase in the polygenic score increased standardized height (adjusted for age, sex and assessment centre and five principal components) by 0.562 SDs ([95% CI 0.558, 0.566], $P < 1 \times 10^{-128}$), equivalent to 5.19cm. Effects were similar in males and females (0.561 SDs [95% CI 0.555, 0.567] and 0.564 SDs [95% CI 0.558, 0.569], respectively).

**Statistical analysis identifies 244 individuals misaligned to genetically predicted height.**   Using a simulated dataset of 158,951 individuals and 3,198 SNPs explaining 31.6% of the variance under an additive model (see methods), we classified 244 individuals of the 158,951 individuals from the UK Biobank as deviating from the polygenic expectation, using Mahalanobis distances based on means of the standardized polygenic scores and adjusted height measures, accounting for covariance between the two variables. Of the individuals deviating from expectation, 150 and 94 individuals were relatively short or tall for their polygenic score, respectively (Fig 2). A total of 109,858 individuals were classified as aligned to their polygenic prediction (residual <1SD) and formed the comparison group for enrichment analyses.

**Observed enrichment of characteristics associated with childhood height, body shape, and rare genetic and non-genetic factors among individuals who deviate from polygenic expectation.**   *Individuals misaligned to their genetically predicted height are more likely to recall being shorter or taller than average at age 10.* As a validation of our polygenic deviation classification for height, we hypothesised that individuals deviating from polygenic expectation were likely to self-report being shorter or taller than average during childhood. We tested for

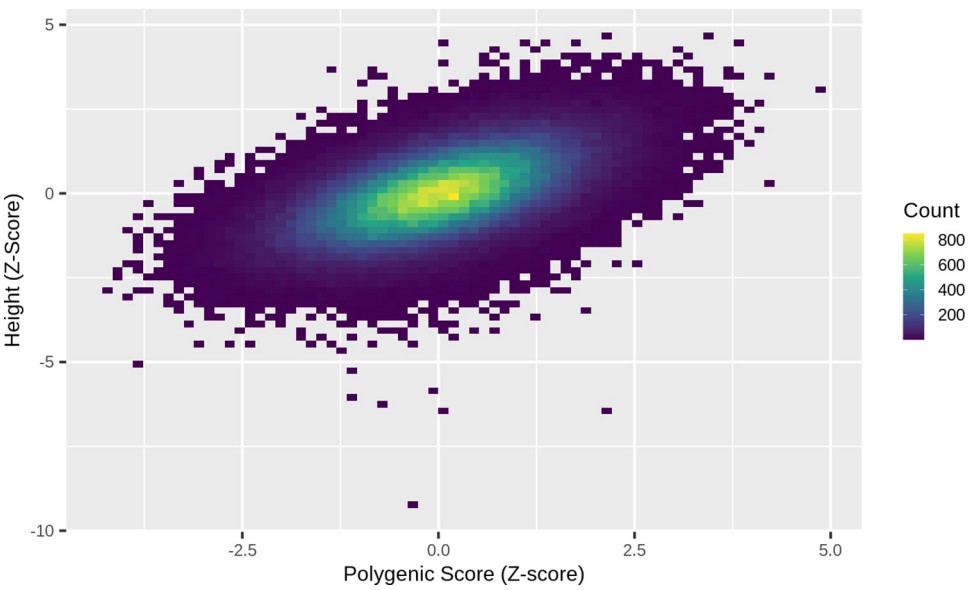

**Fig 1. A density plot of standardized polygenic scores for height plotted against standardized height for 158,951 unrelated individuals from the UK Biobank.**

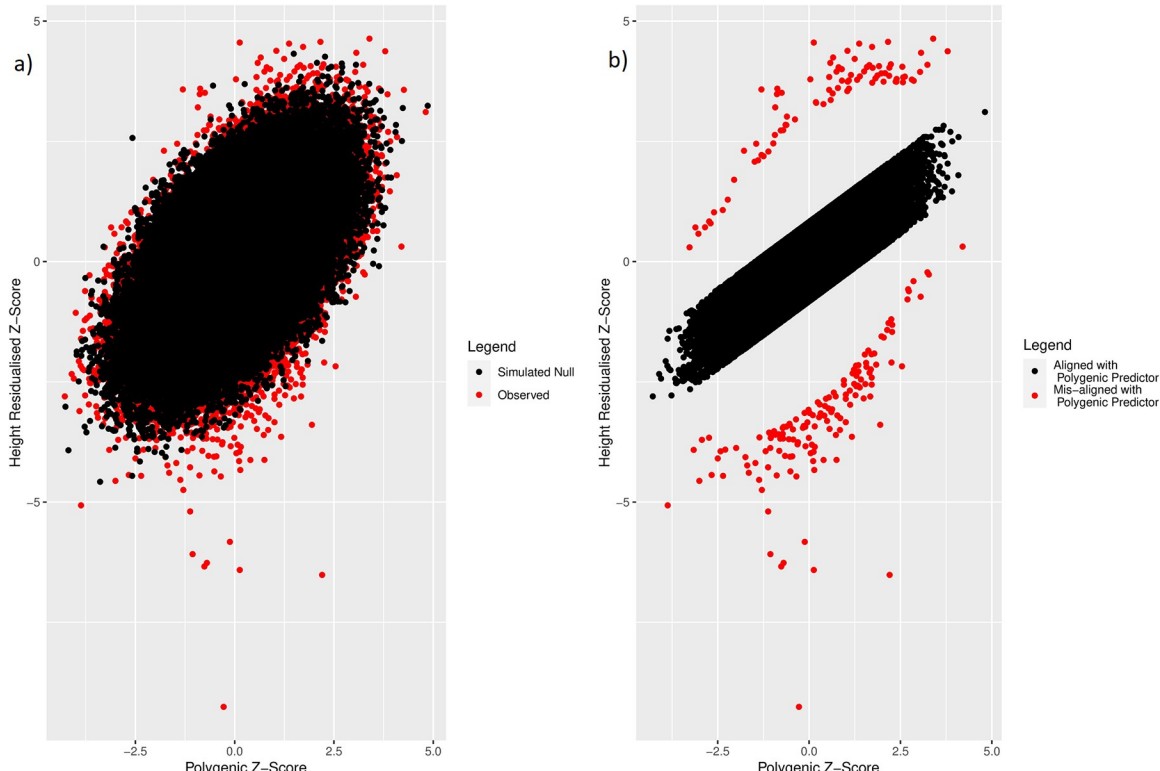

**Fig 2.** a) Observed (red) and simulated (black) polygenic scores and standardized height adjusted for age, sex and assessment centre. b) Individuals aligned (black) and misaligned (red) to genetically predicted height defined using Mahalanobis distance $P < 0.001$ and being more than 2 standard deviations away from the mean of the residual distribution generated by regressing the polygenic score against height. Individuals who were neither classified as aligned or misaligned were removed from b.

enrichment of self-reporting being shorter or taller than average at age 10 among individuals who were shorter or taller than genetically predicted, respectively. We observed evidence of enrichment in both the short and tall deviator groups relative to the group aligned to their genetic score with OR = 10.1 [95% CI 7.19, 14.2], $P = 2 \times 10^{-42}$ and OR = 10.4 [95% CI 6.52, 16.5], $P = 4 \times 10^{-27}$, respectively.

*Individuals who deviate from their genetically predicted height are enriched for having a disproportionate body stature.* As individuals at the extremes of the polygenic score distribution for height are enriched for recalling being shorter or taller at age 10, we next hypothesised that individuals classified as deviating from their genetically predicted phenotype are also more likely to have disproportionate body sizes that affect standing height and have more extreme sitting-to-standing height ratios. We observed individuals who were shorter or taller than genetically predicted were enriched for extreme values of sitting-to-standing height ratio (greater than 1SD) with OR = 2.99 [95% CI 2.12, 4.15], $P = 1.22 \times 10^{-10}$, OR = 6.39 [95% CI 1.72, 53.4], $P = 7.85 \times 10^{-4}$, respectively.

*Individuals with shorter stature than genetically predicted are enriched for congenital malformations and deformations of the musculoskeletal system.* To identify potential reasons why individuals deviate from polygenic prediction, we first tested for enrichment of clinical diagnoses of congenital malformations and deformations of the musculoskeletal system as captured by ICD9 (754–756) and ICD10 (Q75-Q69) codes from Hospital Episode Statistics and primary care data where an ICD9 or ICD10 code could be extracted. We observed an enrichment within the group of individuals with shorter stature misaligned to the genetic predictor with

an odds ratio of 3.45 [95% CI 2.11, 5.65], $P = 2 \times 10^{-5}$ of having a diagnosis of congenital malformations and deformations of the musculoskeletal system but observed a lack of enrichment among the taller group (OR = 1.00 [95% CI 0.999, 1.00], $P = 0.783$).

*Individuals who are shorter relative to their genetically predicted height are enriched for loss-of-function variants in genes most commonly associated with monogenic forms of short stature.* We next hypothesised that individuals classified as having relatively short or tall stature given their polygenic score for height would be enriched for rare variants in dominantly inherited genes previously associated with growth disorders, including overgrowth.

Using 238 genes catalogued in OMIM as causally associated with short or tall stature (see methods) with at least one dominant pattern of inheritance, we first tested whether individuals classified as deviating from polygenic expectation were enriched for any rare (minor allele frequency < 0.1%) loss-of-function (LoF) variants in those genes. We did not observe evidence (at $P < 0.05$) for enrichment of rare LoF variants present in people defined as relatively short for their polygenic prediction (OR = 1.39 [95% CI 1.00, 1.94], $P = 0.071$). However, we did observe a stronger enrichment for LoF carriers when limiting the analysis to a subset of 6 genes (*SHOX*, *NPR2*, *ACAN*, *IGF1*, *IGF1R*, and *FGFR3*) in which variants are known to be relatively common Mendelian causes of short stature (OR = 78.4 [95% CI 40.1, 153.3], $P = 6.83 \times 10^{-16}$) (see methods).

Among individuals with relatively tall stature for their genetic prediction, we did not observe evidence for enrichment of rare LoF variants residing in the 238 genes (OR 1.11 [95% CI 0.699, 1.75] $P = 0.63$). These results were nominally significant ($P < 0.05$) when limiting our analysis to 3 genes in which variants have previously been described as causal for some of the most prevalent syndromes associated with tall stature, specifically Marfan syndrome (*FBN1*) [6–8], Weaver syndrome (*EZH2*) [9], and Sotos syndrome (*NDS1*) [10] (OR = 43.7 [95% CI 1.06, 271], $P = 0.024$).

*Individuals misaligned to their genetically predicted height showed no enrichment of inbreeding.* Following on from previous research that has suggested an association between inbreeding and reduced adult height [11], we next tested whether inbreeding could be associated with our definition of deviation from polygenic expectation. However, we found no evidence of association between the inbreeding F-statistic when comparing individuals who were shorter than genetically predicted versus those who were concordant with their genetically predicted height ($\beta = -0.0488$ [95% CI -0.207, 0.109], $P = 0.54$). We also observed no evidence of association in those who were taller than expected ($\beta = -0.0559$ [95% CI -0.256, 0.144], $P = 0.58$).

*Individuals who are shorter relative to their genetic predictor for height are enriched for lower socioeconomic status.* Finally, we explored whether non-genetic factors could influence whether an individual was classified as deviating from their genetically predicted height given their observed height. Specifically, we assessed the effect of socioeconomic status as represented by the Townsend deprivation index (TDI). We observed an enrichment of higher TDI (representing lower socioeconomic status) among individuals who were relatively short given their genetically predicted height (OR = 2.69 [95% CI 1.92, 3.76], $P = 5.97 \times 10^{-8}$). We did not observe evidence that taller individuals were enriched for lower levels of TDI (OR = 1.122 [95% *CI* 0.625,2.02], P = 0.64).

**Findings remain consistent after applying alternative methods to define individuals deviating from polygenic predictions.** Given our primary analysis was based on using Mahalanobis distances (P<0.001) to define individuals deviating from polygenic predictions, we performed several sensitivity analyses to determine if our overall findings would change if different thresholds and methods were applied to define individuals deviating from polygenic expectation (see methods). Briefly, alternative approaches to define polygenic deviators that assume trait normality included 1) using Mahalanobis distances with $P < 0.05/n$, 2) using

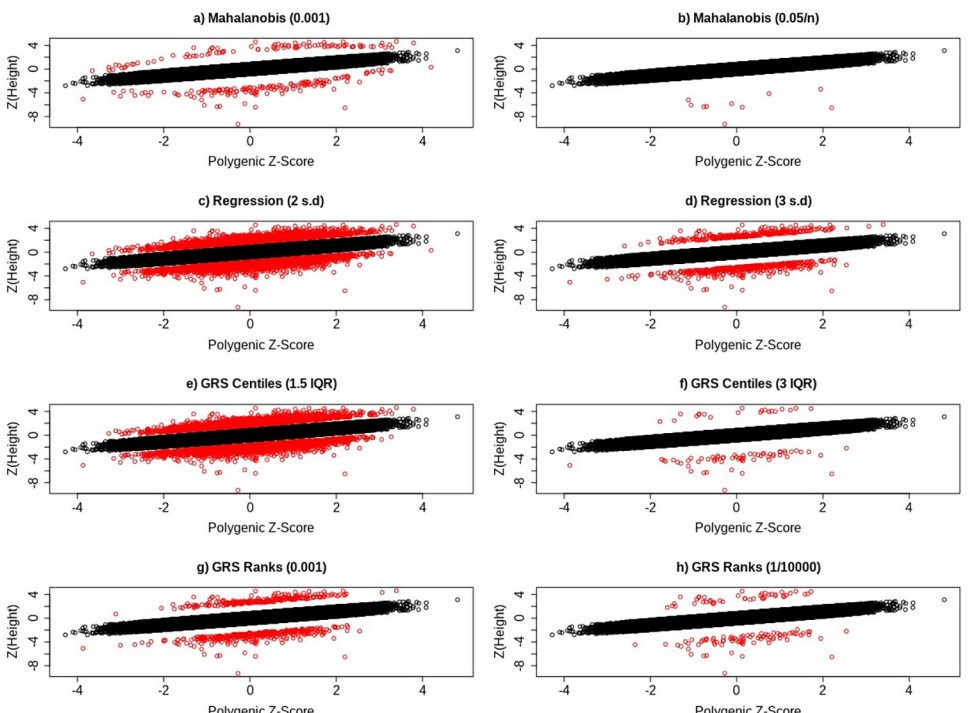

**Fig 3.** Scatter plots showing the distribution of individuals who deviate (red) and do not deviate (black) from their genetic predictor for height, based on a) Mahalanobis distances with $P < 0.001$ and b) $P < 0.05/n$, c) regression residuals at the 2SD and d) 3SD threshold, e) GRS centiles with a Q3 + 1.5 IQR and f) Q3 + 3 IQR threshold, and g) GRS rank with $P < 0.001$ and (h) $P < (1/10000)$.

absolute standardised residual values greater than a) 2 or b) 3 after regressing observed polygenic scores against observed height values, and 3) using empirical P-values based on 10,000 simulations of phenotypes and polygenic score whereby an observed phenotype at a given rank of polygenic score (PS-rank) is compared with 10,000 simulated phenotypes at the same simulated PS-rank. In addition, we implemented a non-parametric centile approach that made no assumptions about the distribution of the quantitative phenotype under examination. While the number and intersection of individuals grouped into the taller and shorter groups differed depending on the method and threshold used (Tables B, C, and D in S1 Text), our findings were largely unchanged (Table E and F in S1 Text). Fig 3 shows how the methods for defining deviator status vary visually.

## LDL cholesterol

**A polygenic score for LDL cholesterol explains 16.7% of the variance in the UK Biobank.** We derived an LDL-C polygenic score for 134,979 unrelated European individuals with measures of LDL-C (UKB Field 30780) and exome-sequencing data in the UK Biobank. We used 1,239,184 SNP effect estimates from the latest meta-analysis of LDL cholesterol (LDL-C) that excluded UK Biobank [4]. The polygenic score explained 16.7% of the variance in LDL-C.

A 1SD increase in the polygenic score increased rank-inverse normalised residualised LDL-C (adjusted for statin use, age, sex and assessment centre and five genetic principal components) by 0.408 SDs ([95% CI 0.403, 0.413], $P < 1 \times 10^{-128}$), equivalent to 0.866 mmol/l. When repeating this analysis in 61,598 males and 73,377 females separately, the polygenic

score explained 16.2% and 18.0% of the variance, respectively. A 1SD change in the polygenic score resulted in a 0.402 SD [95% CI 0.395, 0.409] and 0.424 SD [95% CI 0.417, 0.430] change in LDL-C in the males and females, respectively.

## Statistical analysis classifies 159 individuals as misaligned to their genetically predicted LDL cholesterol

We used the Mahalanobis metric to classify individuals who deviated from their polygenic score. Based on 134,979 individuals and 1,239,184 variants that explained 16.7% of the variance of a normally distributed outcome, we classified 159 individuals from the UK Biobank as deviating from the polygenic expectation (P<0.01), and 92,897 individuals as aligned to their polygenic score (residual < 1SD).

Of those 159 individuals classified as misaligned, 91 and 68 had a relatively low or high LDL-C for their polygenic score, respectively. In a sex stratified analysis, motivated by the sex-heterogeneous nature of lipid levels, 53 and 38 males had relatively low or high LDL-C respectively. Additionally, 41 and 44 females had relatively low or high LDL-C respectively. An additional 17 females were classified as misaligned to their polygenic score in the sex stratified analysis, 14 (82.4%) of which had a higher LDL-C than expected. The absolute number of males classified as misaligned to their polygenic score did not change in the sex-stratified analysis, but the relative number of individuals who had a polygenic score higher than expected increased by 12.1%. Due to these differences, we used the sex-stratified analysis as our primary results. We provide scatter plots in Fig 4 showing how these individuals are distributed as compared to controls, as well as scatter plots showing how this distribution changes for the different methods that we have introduced to classify polygenic misalignment. Counts of polygenic deviators for each method are also given in Table G in S1 Text. A total of 42,652 and 50,461 individuals were classified as aligned to their polygenic prediction (residual <1SD) and formed the comparison group for the following male and female sex-specific enrichment analyses, respectively.

**Observed enrichment of characteristics associated with cardiovascular risk, diabetes and rare genetic factors among individuals who deviate from polygenic expectation.** *Individuals who deviate from their genetically predicted LDL-cholesterol had differing levels of common cardiovascular risk factors.* Our primary hypothesis for the LDL-C phenotype was that individuals whose LDL-C was not aligned with their polygenic prediction would have differing levels of common cardiovascular risk factors. Compared to individuals classified as not deviating from their genetically predicted LDL-C levels, males with high LDL-C relative to their polygenic score had higher triglyceride levels ($\beta$ = 0.695 [95% CI 0.403, 0.985], $P = 2.87 \times 10^{-6}$) and nominally higher HDL levels ($\beta$ = 0.247 [95% CI -0.017, 0.510], $P = 0.0667$). All effect sizes are in sex-specific SD units. Based on the same comparison in females, individuals with a high LDL-C for their polygenic score had higher triglyceride levels ($\beta$ = 0.877 [95% CI 0.635, 1.12], $P = 1.29 \times 10^{-12}$), higher BMI ($\beta$ = 0.636 [95% CI 0.321, 0.950], $P = 7.35 \times 10^{-5}$) and higher cigarette use ($\beta$ = 0.303 [95% CI 0.0838, 0.523], $P = 6.76 \times 10^{-3}$).

Compared to individuals labelled as aligned to the genetically predicted LDL-C, males whose LDL-C was low for their polygenic score had lower triglyceride levels ($\beta$ = −0.885 [95% CI -1.13, -0.638], $P = 2.00 \times 10^{-12}$), lower HDL levels ($\beta$ = −0.632 [95% CI -0.855–0405], $P = 3.00 \times 10^{-8}$) and nominally lower diastolic blood pressure ($\beta$ = −0.271 [95% CI [-0.507, -0.03], $P = 0.0246$). In females, individuals with a low LDL-C for their polygenic score had lower triglyceride levels ($\beta$ = −0.983 [95% CI -1.23, -0.732], $P = 1.64 \times 10^{-14}$) and were nominally older ($\beta$ = 0.353 [95% CI [0.0531, 0.652], $P = 0.0210$)—see Fig 5, Table B in S1 Data and Table H in S1 Text for all Q-risk factors that were assessed.

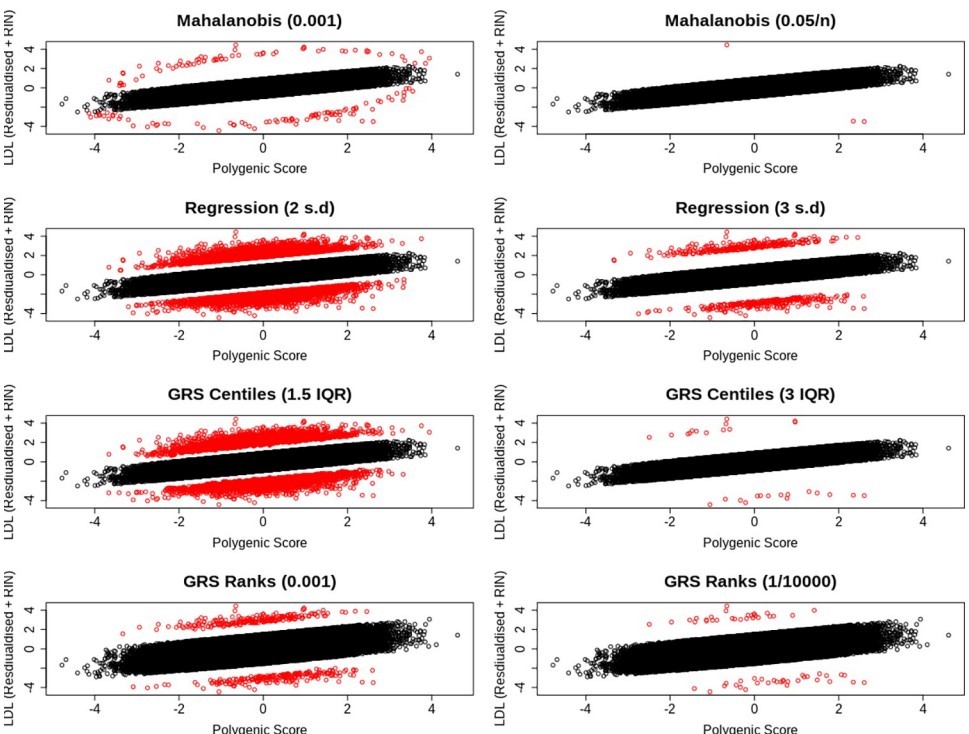

**Fig 4.** Scatter plots showing the distribution of individuals who deviate (red) and do not deviate (black) deviate their genetic predictor for LDL cholesterol, based on a) Mahalanobis distances with $P < 0.001$ and b) $P < 0.05/n$, c) regression residuals at the 2SD and d) 3SD threshold, e) GRS centiles with a Q3 + 1.5 IQR and f) Q3 + 3 IQR threshold, and g) GRS rank with $P < 0.001$ and (h) $P < (1/10000)$.

*Deviation from genetically predicted LDL-C increases the risk of having coronary artery disease and diabetes, even after adjusting for the effects of LDL-C, BMI and age.* Having demonstrated that individuals who deviate from their polygenic prediction had different background risk levels for cardiovascular diseases, we next hypothesised that those individuals would be more likely to be diagnosed with either coronary artery disease or diabetes. Compared to individuals labelled as aligned to genetically predicted LDL-C levels, females whose LDL-C was high for their polygenic score had a nominally increased risk of T2D (OR = 7.07, [95% CI 1.38, 36.2], $P = 0.019$), even after adjusting for the effects of measured LDL-C, age and BMI. We did not observe an association with higher risk of T2D in males labelled as deviating from genetically predicted LDL-C.

Among males classified as misaligned to their LDL-C genetic predictor and whose LDL-C was lower than expected, we observed an enrichment for coronary artery disease (OR = 4.82, [95% CI 2.57, 9.02], $P = 8.87 \times 10^{-7}$) and nominally higher risk of type-two diabetes (OR = 2.32, [95% CI 1.10, 4.90], $P = 0.0278$). In females, individuals with a low LDL-C for their polygenic score showed no evidence of enrichment for T2D or CAD. Refer to Fig 6 and Table B in S1 Data for all results.

*Individuals who deviate from their genetically predicted LDL-cholesterol were more likely to be carriers of damaging exome-sequenced loss-of-function variants in LDLR, APOB and PCSK9.* As considered for height, we finally hypothesised that individuals who were misaligned to the polygenic prediction for their phenotype would be enriched for rare genetic variants in key monogenic genes. Males and females whose LDL-C was high for their LDL-C polygenic score showed evidence of enrichment for rare (< 0.1%) loss-of-function variants in the *LDLR* gene

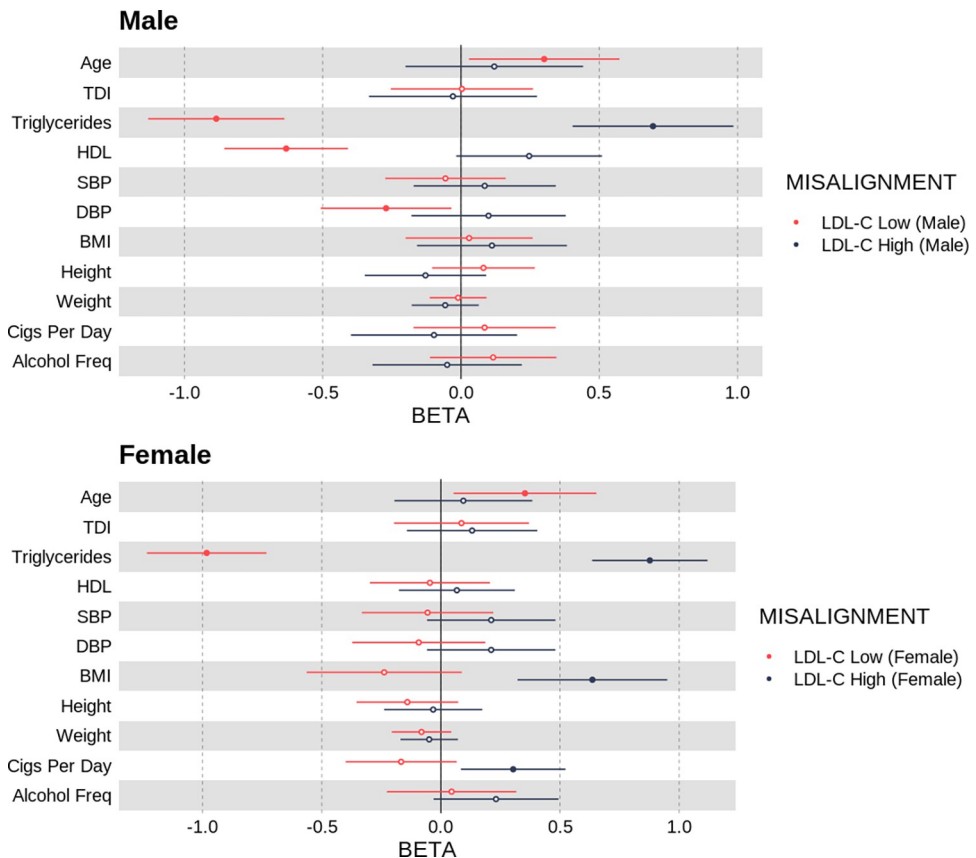

**Fig 5. Odds ratio per standard deviation increase in Q-Risk exposure phenotypes with respect to being classified as a deviating for a polygenic score for LDL cholesterol.**

(males: OR = 4.28 [95% CI 2.28, 8.02], $P = 5.96 \times 10^{-6}$; females: OR = 4.02 [95% CI 2.17, 7.44], $P = 1.02 \times 10^{-5}$).

Males and females whose LDL-C was low for their LDL-C polygenic score showed evidence of enrichment for rare loss-of-function variants in *APOB* (males: OR = 5.49 [95% CI 4.30, 7.02], $P = 4.12 \times 10^{-42}$; females: OR = 5.29 [95% CI 4.11, 6.84], $P = 1.34 \times 10^{-37}$), and for males in *PCSK9* (males: OR = 4.99 [95% CI 3.48, 7.17], $P = 2.54 \times 10^{-18}$).

Refer to Fig 7 and Table B in S1 Data for all exome-sequencing derived enrichment results.

**Using the GRS-ranking method classifies more individuals as deviating from their polygenic LDL-C score, with similar features and some stronger statistical associations.** We additionally classified individuals who were misaligned to their polygenic score for LDL-C using the GRS ranking method, interquartile ranges, and residuals derived from regressing LDL-C on the polygenic score. The results of classifying deviators from a polygenic score for each of the four methods can be found in Tables G and H in S1 Text. Although the number of individuals who were classified as deviating from their polygenic score was 176.1% higher using the GRS-ranking method, the features of those individuals were similar, with the same sign of effect in 73.5% of all analyses. Additionally, with the higher number of individuals classified as deviating, the strength of the statistical association was stronger for some analyses. For example, even after adjusting for BMI, age and measured LDL-C, individuals whose LDL-C was higher than expected based on the GRS-ranking method were much more likely to suffer from type-two diabetes (males: OR = 10.3 [95% CI 3.93, 26.9], $P = 2.09 \times 10^{-6}$). We

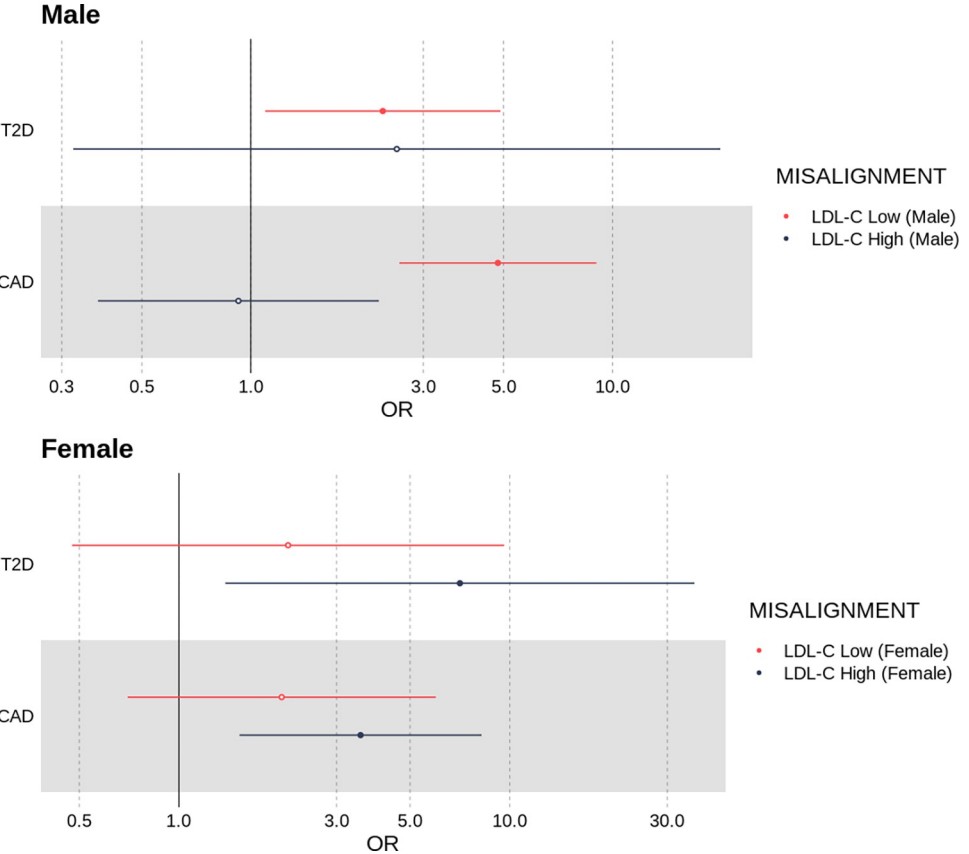

**Fig 6. Odds ratios for an individual having either type two diabetes (T2D) or coronary artery disease if they classified as misaligned to their LDL-C polygenic score, adjusted for BMI, age and LDL-C.**

present all GRS-ranking method results in Tables G and H in S1 Text, alongside those derived from the Mahalanobis method.

## Discussion

We have established novel, robust methods for identifying individuals whose phenotype is misaligned to their polygenic prediction, which we referred to as deviating from a polygenic score, applied to two well-known phenotypes: height, chosen for its high heritability and strongly predictive polygenic score, and LDL-C, chosen for being clinically actionable into adulthood, with a range of associated co-morbidities.

Our results were broadly consistent across the methods tested and are thus likely to be applicable to a range of phenotypes. With ever-increasing sample sizes, we suspect more traits will have highly powered polygenic risk scores that increase the efficacy of this method.

Several lines of evidence indicate that our approach is effective. First, we found, for both standing human height and LDL-C, individuals who deviated from their expected genetic score were enriched for rare genetic mutations in several genes known to be associated with extreme stature and LDL-C. These mutations were discovered using the whole exome sequence data in UK Biobank, and occurred in established genes, such as *ACAN* and *SHOX* for height and *LDLR* and *PCSK9* for LDL-C. These results are similar to that of Lu et. al [12], who found an enrichment of rare damaging variants in individuals with common diseases despite having a low polygenic risk score. Second, individuals who deviated were also enriched

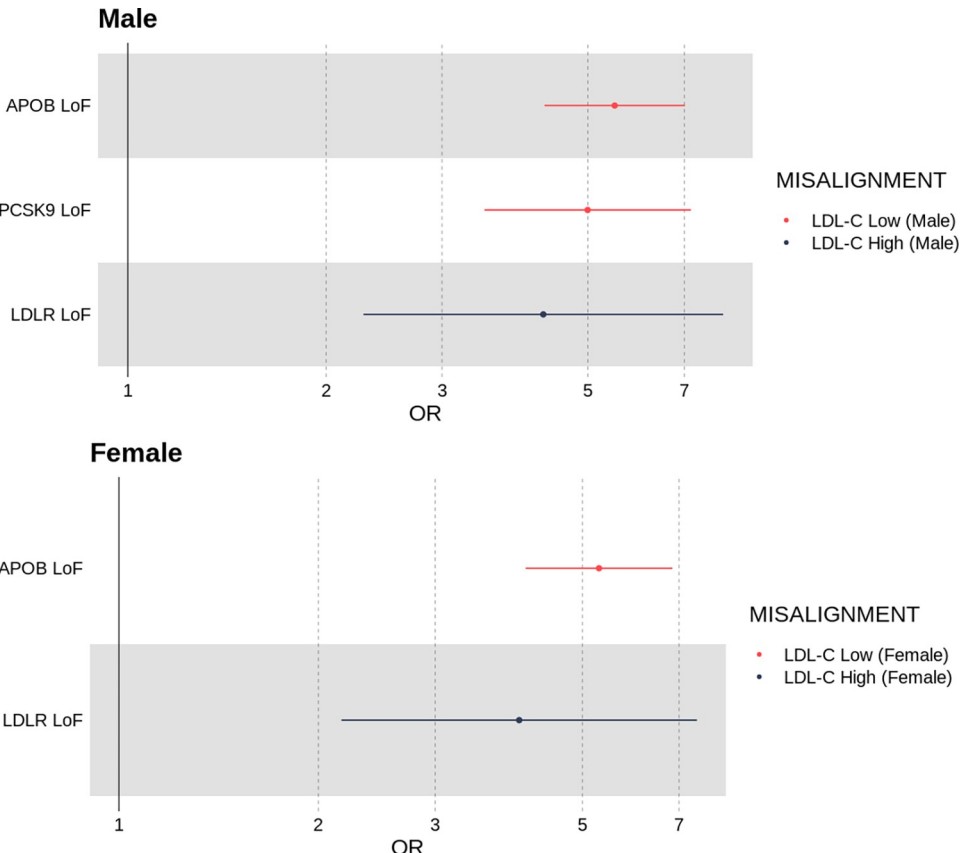

**Fig 7. Odds ratio of an individual being a carrier of a loss-of-function variant in one of three genes known to affect LDL-C levels: (*LDLR*, *APOB* and *PCSK9*) if they were classified as misaligned to their LDL-C polygenic score.**

for other factors known to be associated with differences in phenotype, such as differences in BMI, smoking, and socio-economic position for LDL-C, consistent with the expectation that lifestyle factors play a role in deviation from polygenic prediction. For LDL-C, these differences were also reflected in different risks of heart disease and type 2 diabetes. Other environmental factors not assessed in the manuscript may be drivers for polygenic misalignment and may encompass gene-environment interactions as well.

The number of individuals identified as deviators from their expected phenotype given their polygenic risk varied by method and statistical threshold used. For example, based on the less stringent statistical thresholds (Fig 2A, 2C, 2E and 2G for height) the four methods identified between 244 and 7,316 individuals for height and between 158 and 6,402 individuals for LDL-C. Using the more stringent thresholds (Fig 2B, 2D, 2F and 2H for height) the four methods identified between 10 and 702 individuals for height and between 3 and 577 individuals for LDL-C. Across all Q-risk outcomes, as compared to individuals who had either a lower or higher LDL-C than expected classified using Mahalanobis distance at the standard threshold ($P<0.001$), the statistical evidence for association with Q-risk criteria was stronger ($p<0.05$) when individuals were classified by either the IQR (Q3 + 1.5IQR) or GRS residual ($>$2SD) methods: the two methods which classified the largest number of individuals as misaligned to their polygenic score.

Given both height and the genetic predictor are normally distributed, we were able to use both parametric and non-parametric methods to define individuals who are phenotypically misaligned to their genetic prediction based on the additive model of inheritance. However, phenotypes such as body-mass-index (BMI) are known to be skewed [13] and therefore the non-parametric approaches discussed in this study are more likely to be suitable for other phenotypes analysed on the raw scale and are recommended if rank-based normalisation of the phenotype, for example, is not implemented.

There are some limitations of this study. First, while the primary method is suited for normally distributed phenotypes and genetic scores, as observed for height, no optimal Mahalanobis distance threshold is known. We have attempted to overcome this by demonstrating the efficacy of our method on LDL-C, a skewed phenotype. We have also shown that our results remain largely consistent when changing statistical thresholds that guide inclusion of individuals to follow-up who are deviating from polygenic expectation. Second, the UK Biobank is healthier than the general population [14], which may have affected our ability to identify people with rare genetic or non-genetic causes to their phenotype. Third, the methods applied, and analysis performed in this manuscript rely on polygenic scores that explain an appreciable amount of variance in a trait. However, even the most predictive polygenic risk scores to date are known to exhibit uncertainty [15] that could affect our proposed GRS-ranking methodology. Furthermore, the utility of this work in under-represented populations in GWAS studies is likely to be more limited presently. Our work has focussed on individuals of European genetic ancestry: recent work has shown how the continuum of genetic ancestry can impact the predictability of a polygenic score [3,16]. In addition, our approach has not been applied to groups of individuals from more heterogeneous populations. More work is required to develop and evaluate existing statistical approaches to identify individuals likely deviating from polygenic expectation from such populations. Fourth, a potential explanation for polygenic deviation is sample mix-up [17]. However, we did not apply methods to determine this because 1) some methods rely on phenotypic mismatch with polygenic expectation, and 2) we have used samples that have not been flagged by UK Biobank has having sex-mismatches. Importantly, we did not observe an overlap between individuals classified as misaligned to the height and LDL-C polygenic scores that may be indicative of sample mix-up if missed by sex-checks. Finally, we note that analysis of socioeconomic status during adulthood may not necessarily serve as a good proxy for socioeconomic status at childhood during the key stages of growth and development when the living environment has the potential to act adversely on growth. In addition, we note that genetics can determine socioeconomic status [18] and is not strictly a measure of the effect of an individual's environment.

In conclusion, our results support the hypothesis that individuals who deviate from their genetically predicted phenotype, as defined by common variants and using a suite of statistical methods, are of clinical interest. These individuals are more likely to carry rare genetic variation, or be at greater risk of co-morbidities, and should be considered in future discovery studies.

## Methods

### Ethics statement

The UK Biobank was granted ethical approval by the North West Multi-centre Research Ethics Committee (MREC) to collect and distribute data and samples from the participants (http://www.ukbiobank.ac.uk/ethics/) and covers the work in this study, which was performed under UK Biobank application numbers 9072. All participants included in these analyses gave written consent to participate.

## Study population

We analysed 158,951 unrelated individuals from the UK Biobank with inferred European genetic ancestry as previously described [19]. All individuals had measurements for height, genetic data derived from genome-wide array-based imputation, and whole-exome sequence data, as described in [20]. Of those 158,951 individuals, 134,979 also had measure of LDL cholesterol from blood biochemistry.

## Phenotypic derivation

Height (cm) was derived from the UK Biobank (field 50) and converted to standardized residuals, after adjustment for age, sex and UK Biobank assessment centre. We subsequently defined short/tall stature as a residualised height > 2 standard deviations from the mean.

LDL cholesterol (mmol/l) was derived from the UK Biobank (field 30780) and converted to rank-inverse normalised residuals, after adjustment for medication, age, sex and UK Biobank assessment centre.

## Derivation of a polygenic predictor for height

We created a genetic predictor for height (Eq (1)) for each of the unrelated 158,951 individuals using conditional effect estimates of 3,198 SNPs reaching $P \leq 5 \times 10^{-8}$ from an interim meta-analysis of height performed by the Genetic Investigation of Anthropometric Traits (GIANT) consortium in up to 1,400,860 individuals (mean N = 1,148,694) that excluded the UK Biobank (Table C in S1 Data).

We created a genetic predictor for LDL-C (Eq (1)) for each of the unrelated 134,979 individuals using PRS-Cs [21] applied to GWAS summary statistics of 1,239,184 SNPs from [4], based on an interim analysis that excluded UK Biobank.

We calculated the genetic predictors using the following formula:

$$PS_i = \sum \beta_j G_j \tag{1}$$

where $PS_i$ refers to the $i^{th}$ individual's polygenic score, calculated as the overall sum of the effect sizes of each SNP $j$ ($\beta j$) multiplied by an individual's genotype for the respective SNP ($G_j$). The genetic predictors were subsequently corrected for the first five principal components, calculated within a broader set of unrelated European individuals from the UK Biobank [22]. Finally, the distribution of the genetic predictors adjusted for genetic ancestry were standardized with $\mu = 0$ and $\sigma = 1$.

## Identifying individuals who deviate from their expected phenotype

For our primary analysis on standing height, we defined two statistical criteria for labelling individuals as deviating from their expected height given their genetic height score. First, we estimated the variance explained by the genetic predictor in the 158,951 individuals from the UK Biobank. Next, we simulated 158,951 individuals and 3,198 SNPs under the additive polygenic model whereby the phenotypic variance explained by the simulated SNP effects approximated those observed in the UK Biobank. We subsequently calculated a polygenic score for each simulated individual (Eq (1)) prior to deriving the covariance matrix of the standardized simulated phenotypes and standardized polygenic scores. Next, we calculated Mahalanobis distances for the standardized observed height measures and polygenic scores using the covariance matrix from the simulated dataset. All Mahalanobis distances were subsequently converted to P-values based on a $\chi^2$ distribution with 2 degrees of freedom to represent the probability of a data point being an outlier relative to the correlation between the genetic

predictor and observed phenotype. We used P-value thresholds of $< 0.001$ to define individuals deviating from their expected phenotype.

Second, to account for the possibility of outlying Mahalanobis distances being associated with individuals with both an extreme polygenic score and height measurement, consistent with the additive polygenic model, we regressed the observed standardized polygenic scores against the observed standardized heights and retained individuals reaching our P-value threshold if $|z| > 2$, where $z$ represents the z-score of the normalised residuals of the regression model. Individuals with $|z| < 1$ were defined as being consistent with the additive polygenic model.

Individuals classified as deviating from their expected phenotype were subsequently split into two groups dependent on whether their standardized height was below the mean (shorter) or above the mean (taller) for follow-up analyses.

## Testing for enrichment of characteristics among individuals deviating from genetically predicted height

We performed separate enrichment analysis of several characteristics in the shorter and taller than predicted for their genetically predicted phenotype individuals defined above.

**Self-reporting of being shorter or taller than average at age 10 and sitting to standing height ratio.**   We tested whether individuals who were classified as deviating from the polygenic risk score were enriched for physical observations we may expect. This included self-reporting of being shorter or taller at age 10 (UK Biobank field 1697), and extreme values of the ratio of their sitting-to-standing height ratio (UK Biobank data fields 20015 and 50) adjusted for age, sex and centre.

**Congenital malformations and deformations of the musculoskeletal system defined using ICD9&10 codes.**   To identify individuals previously clinically diagnosed as having congenital malformations affecting the musculoskeletal system we used ICD9 and ICD10 codes available from Hospital Episode Statistics (HES), and primary care data where read codes could be converted to ICD9 or ICD10 codes. We selected ICD9 codes 754–756 (UK Biobank data fields 41203, 41205) and ICD10 codes Q65-Q79 (UK Biobank data fields 41202, 41204) (and the sub-classifications of these codes).

**Rare variants in genes with dominant inheritance catalogued in OMIM as associated with stature phenotypes.**   Using whole-exome sequence data available in the UK Biobank, we tested for enrichment of rare (MAF $< 0.001$) loss-of-function variants residing in a curated list of genes related to short and tall stature from OMIM (Online Mendelian Inheritance in Man) [23]. This list was generated from all genes published in [24] (curated from OMIM queries for short stature, tall stature, overgrowth, brachydactyly, or skeletal dysplasia), plus curated genes from the union of the list in [25] with OMIM queries for short stature in 2019 and 2020, as well as OMIM queries for tall stature, overgrowth, brachydactyly or skeletal dysplasia in 2020, and Endotext skeletal disorders. Specific skeletal phenotypes can be found in S1 Text. From this query, we restricted analysis to a list of 238 genes for which OMIM had catalogued as having at least one dominant inheritance pattern (Table A in S1 Text). Based on the canonical transcripts of the 238 genes, we used VEP [26] and the LOFTEE plugin [27] to annotate variants as loss-of-function with high confidence. We also separately assessed a subset of 6 genes (*SHOX*, *NPR2*, *ACAN*, *IGF1*, *IGF1R*, and *FGFR3*) [28] and 3 genes (*FBN1*, *EZH2* and *NSD1*) [6–10] established as common Mendelian causes of short and tall stature, respectively.

**Inbreeding coefficients.**   It has previously been shown that enhanced inbreeding can lead to lower height [29].

We thus assessed whether the F-statistic for inbreeding was significantly different for those individuals classified as deviating. The F-statistic for inbreeding was calculated using PLINK (v1.9) [30].

**A proxy measure of socioeconomic status.** We tested for enrichment of socio-economic status using townsend deprivation index (UK Biobank data field 189), to determine whether individuals who were short/tall had a depleted/enriched socio-economic status respectively.

## Sensitivity analyses

To determine whether our findings for standing height were based on our primary definition of deviation from polygenic expectation would be generalisable to other definitions, we repeated our analysis using additional statistical thresholds and methods. These included a more stringent Mahalanobis distance threshold of P< 0.05/$n$, where $n$ is the number of individuals in the analysis. As a second approach, we generated standardized residuals for height by regressing the polygenic score for height on height measures and subsequently labelling individuals as deviating from genetic predictions if their ‖z-score‖ was >2 or >3 ('Regression'–Table B in S1 Text). A third approach combined observed data with simulated data. First, each individual was ranked according to their height PS and the corresponding phenotypic values stored. Next, we simulated 158,951 individuals and 3,198 genetic variants matched on the observed allele frequencies and variances explained. Subsequently, a PS was generated for each simulated individual, ranked, and their corresponding phenotype stored. This was repeated 10,000 times. Finally, at each PS rank based on the observed data, we compared the observed phenotype associated with the PS rank with the 10,000 simulated phenotypic values associated with the simulated PS rankings. An empirical p-value was calculated as $(r + 1)/10001$, where $r$ represented the number of simulated phenotypes that were as extreme as that observed at the given PS rank. ('GRS Ranks'–Table B in S1 Text). Finally, we used a non-parametric approach that made no assumption about the distributions of the phenotype or polygenic scores. Specifically, within each centile of the polygenic score, we defined phenotypic outliers as those outside 1) Q1-1.5×IQR to Q3+1.5×IQR (Inter Quartile Range) and 2) Q1-3×IQR to Q3+3×IQR of the standardized height measure, where Q1 and Q3 are the 25th and 75th centiles of the observed height distribution within the GRS centile ('GRS Centiles'–Table B in S1 Text).

## Identifying individuals who deviate from their expected LDL-C

We next identified individuals whose LDL-C was higher or lower than predicted by a polygenic score, again using the Mahalanobis distance as a measure of deviation from polygenic score. The distribution of LDL-C is right-skewed, and as such we applied the GRS-ranking method as a sensitivity analysis because of its less restrictive parameterisation assumptions. We additionally performed a stratified analysis of males and females separately for LDL-C due it being a static measure influenced by sex-heterogenous effects, and the associated differing downstream risk of related outcomes such as coronary artery disease. To maximise the normality of the distributions considered, we rank-inverse normalised LDL-C distributions for each sex independently.

## Testing for enrichment of characteristics among individuals deviating from genetically predicted LDL-C

We performed separate enrichment analysis of several characteristics in the higher LDL-C and lower LDL-C than predicted for their genetically predicted phenotype individuals defined above.

**Cardiovascular Q-Risk phenotypes and disease.** Individuals in the UK who are thought to be at risk of cardiovascular complications are measured on a QRISK scale [31]. The QRISK model accounts for phenotypes such as sex, ethnicity, ancestry, economic deprivation etc. We tested whether individuals who deviated from their polygenic score for LDL-C had higher/ lower (as appropriate) QRISK factors. For a complete list of Q-risk factors tested, and the UKB fields from which they were derived, see Table H in S1 Text. For each QRISK factor we performed a linear regression with the LDL-C misalignment (higher or lower) as an exposure, corrected for sex, UKB assessment centre, age and BMI, excluding when those factors were outcomes. The QRISK outcomes were additionally rank inverse normalised so that effect sizes were scaled by the standard deviation. For downstream risk factors (diabetes, type 2 diabetes and coronary artery disease), we performed a logistic regression where LDL-C misalignment was a risk factor to one of the three outcomes.

**Rare variants in genes with established associations with LDL-C.** Using whole-exome sequence data available in the UK Biobank, we tested for enrichment of rare (MAF < 0.001) loss-of-function variants in one of three genes known to affect levels of LDL-C: *LDLR*, *APOB* and *PCSK9*, as in [32]. As for height, based on the canonical transcripts of the 3 genes, the LOFTEE plugin to annotate variants as loss-of-function with high confidence within VEP.

## Supporting information

**S1 Text. Phenotypic criteria for filtering genes catalogued in OMIM and described as causal for syndromes associated with stature. Table A.** List of 238 genes with prior evidence for a causal association with syndromes associated with stature, filtered on those with evidence of a dominant inheritance relationship. **Table B.** The number of individuals who are defined as deviating from their polygenic score for height using different methodologies, split by those relatively tall and relatively short for their polygenic score (total n = 158,951). **Table C.** Percentage overlap of individuals classified as shorter than expected for their polygenic score for height across derivation methods **Table D.** Percentage overlap of individuals classified as taller than expected for their polygenic score for height across derivation methods. Note, no individuals were classified as being relatively tall when using a Mahalanobis-based P-value threshold = 0.05/n. **Table E.** Empirical P-values for enrichment in individuals who are short relative to their genetically predicted height across all deviator definitions. **Table F.** Empirical P-values for enrichment in individuals who are tall relative to their genetically predicted height across all deviator definitions. No individuals were classified as being relatively tall when using a Mahalanbobis-based P-value threshold = 0.05/n. **Table G.** Number of individuals, and percentage of population, identified as deviating from their polygenic score for measured LDL using different methodologies. **Table H.** UKB Fields used to derive Q-risk measures. (PDF)

**S1 Data. Table A.** Continuous Q-risk outcome regression results for LDL-C polygenic deviators, for all methods. **Table B.** Binary outcome regression results for LDL-C polygenic deviators, for all methods. Analyses where the logistic regression model did not converge are labelled with "NA". **Table C.** SNP weights used to calculate the polygenic score for height (GIANT meta-analysis excluding UKB and 23&Me). (XLSX)

## Acknowledgments

The authors would like to acknowledge the use of the University of Exeter High-Performance Computing (HPC) facility in carrying out this work. We also acknowledge use of high-

performance computing funded by an MRC Clinical Research Infrastructure award (MRC Grant: MR/M008924/1).

## Disclaimer

This communication reflects the author's view: neither IMI nor the European Union, EFPIA, or any Associated Partners are responsible for any use that may be made of the information contained therein. The views expressed are those of the author(s) and not necessarily those of the NIHR or the Department of Health and Social Care.

## Author Contributions

**Conceptualization:** Gareth Hawkes, Michael N. Weedon, Timothy M. Frayling, Andrew R. Wood.

**Data curation:** Gareth Hawkes, Loic Yengo, Sailaja Vedantam, Eirini Marouli, Robin N. Beaumont, Jessica Tyrrell, Michael N. Weedon, Joel Hirschhorn, Timothy M. Frayling, Andrew R. Wood.

**Formal analysis:** Gareth Hawkes.

**Funding acquisition:** Timothy M. Frayling.

**Investigation:** Gareth Hawkes.

**Methodology:** Gareth Hawkes, Andrew R. Wood.

**Project administration:** Timothy M. Frayling, Andrew R. Wood.

**Supervision:** Timothy M. Frayling, Andrew R. Wood.

**Visualization:** Gareth Hawkes.

**Writing – original draft:** Gareth Hawkes, Timothy M. Frayling, Andrew R. Wood.

**Writing – review & editing:** Gareth Hawkes, Loic Yengo, Sailaja Vedantam, Eirini Marouli, Robin N. Beaumont, Jessica Tyrrell, Michael N. Weedon, Joel Hirschhorn, Timothy M. Frayling, Andrew R. Wood.

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
