## [Decision Letter · Decision Letter 0]

24 Apr 2023

Dear Dr Wood,

Thank you very much for submitting your Research Article entitled 'Identification and analysis of individuals who deviate from their genetically-predicted phenotype' to PLOS Genetics.

The manuscript was fully evaluated at the editorial level and by independent peer reviewers. The reviewers appreciated the attention to an important problem, but raised some concerns about the current manuscript. Based on the reviews, we will not be able to accept this version of the manuscript, but we would be willing to review a revised version. We cannot, of course, promise publication at that time.

If you decide to revise the manuscript for further consideration at PLOS Genetics, please aim to resubmit within the next 60 days, unless it will take extra time to address the concerns of the reviewers, in which case we would appreciate an expected resubmission date by email to plosgenetics@plos.org.

We are sorry that we cannot be more positive about your manuscript at this stage. Please do not hesitate to contact us if you have any concerns or questions.

Yours sincerely,

Heather J Cordell

Academic Editor

PLOS Genetics

Hua Tang

Section Editor

PLOS Genetics

Reviewer's Responses to Questions

**Comments to the Authors:**

Reviewer #1: I love the idea of this paper. Identifying the "outliers from predicted values" and starting to characterize why they are outliers is just great and has important clinical and basic science implications (both). It is also an idea that I would expect to capture the public imagination. Overall the paper is well written and clear and the methods are solid. The choice of traits to look at is perfect. I have two concerns, however - one scientific and the other regarding the presentation.

The presentation issue is just that it its a bit disjointed. Each individual analysis is presented clearly, but there's not a narrative of what questions you are asking and why. A few more "topic sentences" at the start of each results section saying how the analysis fits into the big picture of what you're trying to understand about these people would help. Or perhaps more of an introduction to the questions you are going to ask.

My scientific issue/question is very important. Almost all of the analyses are about this group of outlier people being "enriched" for one thing or another, but there is no discussion of "enriched" compared to whom. Is it everyone else? Everyone else with matching height? Something else? For example, saying that these people are "enriched" for having been concerned about height/growth at age 10 seems confusing. Aren't these people short and tall on average compared to the whole population, so wouldn't we expect that? Or are they enriched compared to height-matched individuals who are not outliers relative to prediction? This issue is relevant to pretty much every analysis. At a minimum the comparison groups need to be explained better and the results interpreted for whether they are surprising or what they tell us. Ideally each enrichment analysis uses an appropriate comparison group so that the questions and analyses are as meaningful as possible. One additional data presentation that would be useful is a histogram of the trait value(s) for the outlier people. Are they mostly short? Mostly tall? Uniform across the whole range? (Certainly a person can be predicted short or tall and then actually end up at medium height, but it's not clear whether that would be deviant enough to land them in the outlier group.

Reviewer #2: Hawkes et al. have embarked in characterizing individuals with polygenic misalignment and investigate likely explanations using two highly polygenic traits like body height (with normal trait distribution) and cholesterol (phenotypic skewed), in individuals from the UKBIOBANK with whole-exome data and trait information (n=158,951). The authors identify 0.15% for height and 0.12% with LDL-cholesterol of individuals presenting polygenic misalignment. Further, strong associations with childhood trait presentation, disproportionate stature, diagnosis of growth syndrome, carriership of rare and LOF variants and low socioeconomic status (environmental contribution) was identified for polygenic misaligned individuals and body height. Polygenic misaligned individuals for cholesterol have associations with higher risk of cardiovascular disease and diabetes and for carriership of rare variants in LDL-C.

The study is elegantly conceived and carried out with robust methodology to address the research question. Rather than novel insight this work is a proof-of-concept demonstration useful for the interpretation of genetically-predicted traits, i.e., individuals with polygenic misalignment are more likely to carry rare genetic variation or exposed to environmental factor(s) explaining the difference between predicted and observed trait levels.

1. Can the authors discuss how the uncertainty in PRS estimates at the individual level (see https://doi.org/10.1038/s41588-021-00961-5) affects the interpretation of their findings? Consistent classification as discordant and reproducibility across the different approach employed does not necessarily mean the "true" rank in the PRS has been determined for these individuals. This can also be an explanation why so large variation was observed according to method and threshold classification used. Applying smoothing methods and error-correction methods could provide more certainty of the PRS rank of individuals.

2. A yet not considered explanation for the discordance can be that of undetected sample swaps or mixups… (see https://doi.org/10.1093/bioinformatics/btab783). Likely the discordant samples of the LDL cholesterol and height efforts are not the same or the authors would have reported that. Would be could to include this connotation and succinctly justify why this is not the case here i.e., the possibility of samples swaps has been addressed.

3. It would be informative to provide a map of the polygenic misalignment for the identified discordant individuals. Eyeballing the scatterplots, none of the individuals deviating from the polygenic expectation arise from the extremes of PRS distribution. Rather, most are located in the middle of the plot (PRS distribution), i.e., individuals have an average number of trait- increasing | decreasing alleles but then have higher or lower phenotypic values than expected for the mean of the population. When that is the case, monogenic mutations of large effects are the most likely explanation for the deviation. In absence of mutation, an influential environmental contribution would be expected (i.e., in the middle of the PRS distribution the additive polygenic contribution is averaged out resulting in heritability ~ 0). If the PRS distribution was divided into say quintiles, examine a table of clinical characteristics (like the one used in Figure 5) stratified by polygenic misalignment (yes/no). One would expect randomization of the mean value of the clinical characteristics across PRS ntiles only for the stratum without polygenic misalignment, is that observed? This analysis could help pick up influential environmental factors that are on average not randomized across ntiles in the stratum of individuals with polygenic misalignment.

Minor points

4. How to interpret the association in individuals with polygenic misalignment and self-reporting height at age 10 years and the disproportionate proportion? Is this an environmental contribution (i.e., growth spurt, catch-up, socioeconomic status, etc) or is it related to the effect of specific genetic factors (i.e., SHOX, NPR2, ACAN, IGF1, IGF1R, FGFR3, FBN1, EZH2 and NSD1 have also been found associated with sitting height).

5. Similarly, this is applicable for the higher risk of cardiovascular disease and diabetes association in individuals with polygenic misalignment of LDL-C levels; is this pointing out to an environmental contribution of the influence of life-style and overall health status?

6. Using the previous two points as example, authors can discuss if there is a role for gene-environment interactions in the polygenic misalignment.

7. Given that the likelihood of polygenic misalignment seems to be much less in the extremes of the PRS distribution, the authors can suggest from their work this can be an advantage for risk classification purposes i.e., the comparison of PRS extremes is better suited to evaluate unconfounded contrasts between phenotypes (e.g., Recall by Genotype). Also, support for the use of MR approaches.

8. Line 352 pg 14 “This included self-reporting of bring shorter…” => being shorter

9. Supp Table 5 misses in the footer spelling out the TDI abbreviation.

Reviewer #3: The review is uploaded as an attachment.

**Have all data underlying the figures and results presented in the manuscript been provided?**

Reviewer #1: **No: **UK biobank data has very specific access rules.

Reviewer #2: Yes

Reviewer #3: Yes

PLOS authors have the option to publish the peer review history of their article (what does this mean?). If published, this will include your full peer review and any attached files.

Reviewer #1: No

Reviewer #2: **Yes: **Fernando Rivadeneira

Reviewer #3: No

---

## [Decision Letter · Decision Letter 1]

5 Jul 2023

Dear Dr Wood,

Thank you very much for submitting your Research Article entitled 'Identification and analysis of individuals who deviate from their genetically-predicted phenotype' to PLOS Genetics.

The manuscript was fully evaluated at the editorial level and by independent peer reviewers. The reviewers appreciated the revisions made but identified some remaining minor concerns that we ask you address in a further revised manuscript.

We therefore ask you to modify the manuscript according to the review recommendations. Your revisions should address the specific points made by each reviewer.

Yours sincerely,

Heather J Cordell

Academic Editor

PLOS Genetics

Hua Tang

Section Editor

PLOS Genetics

Reviewer's Responses to Questions

**Comments to the Authors:**

Reviewer #1: The authors did a nice job of responding to both reviewers and clarifying a number of things that were unclear in the original submission. I still like the univariate histograms better than the scatterplots, but that's OK. I defer to how the authors want to present their data.

Reviewer #2: I thank the authors for the effort placed on making a thorough revision of their paper which address well the reviewer comments.

Yet, I consider my second point (i.e., possibility of sample swaps explaining the polygenic deviation) requires minor revision, plus one consideration about the scope of the findings:

1. The text in the discussion section of the manuscript (pg 13/21 lines 313-316) is incomplete and with a typo: "Fourth, a potential explanation for polygenic deviation is sample mix-up and <what?>[17]. However, we did not apply methods to determine this because 1) some methods rely on phenotypic mismatch with polygenic expectation, and 2) we have used samples that have not been flagged by UK Biobank <h>as having sex-mismatches."

2. In relation to the previous point, the question "whether the reported polygenic deviations consequence of sample swaps or not?", is not fully answered. While cross-sex sample swaps may have been identified, same-sex swaps will run undetected and the numbers (0.15% for height and 0.12% for LDL-C) seem consistent with those few who may have escaped the QC control. Rather, specify in the discussion that the deviants in the height analysis are NOT the same individuals of the LDL-C analysis; just as Idefix works by identifying those individuals that behave as deviants across multiple PRS's.

3) The table provided for my raised point 3 with subject characteristics across LDL-C quintiles (in women with lower measued LDL-C) lacks the sample size (n) of deviators per quintile. I could predict, Q4 is where the largest amount of deviant is given that it holds the biggest contrast in age is observed, correct?. Strikingly, all deviants (independent of quintile) show rather consistently lower BMI, alcohol use and smoking. This is somewhere marginally stated by the authors along the manuscript. Yet, this also means a healthy lifestyle "overrides" the genetic predisposition... don't you want to emphasise more that message? In the conclusion, authors focus only on one perspective: "These individuals are more likely to carry rare genetic variation, or be at greater risk of co-morbidities, and should be considered in future discovery studies." You can also emphasise such positive perspective that has great potential to catch media attention. ;)</h></what

Reviewer #3: All of my comments have been nicely addressed by the authors. I have no further suggestions to make.

**Have all data underlying the figures and results presented in the manuscript been provided?**

Reviewer #1: Yes

Reviewer #2: Yes

Reviewer #3: Yes

PLOS authors have the option to publish the peer review history of their article (what does this mean?). If published, this will include your full peer review and any attached files.

Reviewer #1: No

Reviewer #2: **Yes: **Fernando Rivadeneira

Reviewer #3: No

---

## [Decision Letter · Decision Letter 2]

22 Aug 2023

Dear Dr Wood,

We are pleased to inform you that your manuscript entitled "Identification and analysis of individuals who deviate from their genetically-predicted phenotype" has been editorially accepted for publication in PLOS Genetics. Congratulations!

Yours sincerely,

Heather J Cordell

Academic Editor

PLOS Genetics

Hua Tang

Section Editor

PLOS Genetics

Comments from the reviewers (if applicable):

Reviewer's Responses to Questions

**Comments to the Authors:**

Reviewer #2: All my comments have been adequately addressed.

**Have all data underlying the figures and results presented in the manuscript been provided?**

Reviewer #2: Yes

PLOS authors have the option to publish the peer review history of their article (what does this mean?). If published, this will include your full peer review and any attached files.

Reviewer #2: **Yes: **Fernando Rivadeneira

**Data Deposition**

http://datadryad.org/submit?journalID=pgenetics&manu=PGENETICS-D-23-00163R2

**Press Queries**

---

## [Editor Report · Acceptance letter]

18 Sep 2023

PGENETICS-D-23-00163R2 

Identification and analysis of individuals who deviate from their genetically-predicted phenotype 

Dear Dr Wood, 

We are pleased to inform you that your manuscript entitled "Identification and analysis of individuals who deviate from their genetically-predicted phenotype" has been formally accepted for publication in PLOS Genetics! Your manuscript is now with our production department and you will be notified of the publication date in due course.

With kind regards,

Anita Estes

PLOS Genetics

On behalf of:
